# Culture and Behaviour Management of Children in the Dental Clinic: A Scoping Review

**DOI:** 10.3390/dj13050186

**Published:** 2025-04-24

**Authors:** Adebola Oluyemisi Ehizele, Love Bukola Ayamolowo, Adeyinka Ishola, Moréniké Oluwátóyìn Foláyan

**Affiliations:** 1Department of Periodontics, School of Dentistry, College of Medical Sciences, University of Benin, Benin City 300001, Nigeria; adebola.ehizele@uniben.edu; 2Department of Nursing Science, Faculty of Basic Medical Sciences, College of Health Sciences, Obafemi Awolowo University, Ile-Ife 220001, Nigeria; layamolowo@oauife.edu.ng; 3Faculty of Nursing, College of Medicine, University of Ibadan, Ibadan 110115, Nigeria; 4Department of Child Dental Health, Faculty of Dentistry, College of Health Sciences, Obafemi Awolowo University, Ile-Ife 220001, Nigeria

**Keywords:** cultural sensitivity, behaviour management techniques, paediatric dentistry, communication, parenting styles, stoicism

## Abstract

Cultural norms, beliefs, and practices influence parental expectations, children’s responses, and the acceptance of behaviour management techniques (BMTs) in paediatric dentistry. Despite this, the existing guidelines often adopt a standardized approach, overlooking critical cultural differences. This scoping review maps the links between culture and behaviour management strategies in paediatric dental settings. A scoping review following PRISMA guidelines was conducted across PubMed, Cochrane Library, Web of Science, Google Scholar, and hand-searched sources from the inception of the databases to 31 January 2025. A total of 671 studies were identified, with 15 meeting the inclusion criteria. Data on the key findings were inductively analyzed to assess cultural influences on parental acceptance, child behavior, and communication. The findings show that non-invasive BMTs such as TellShow–Do and positive reinforcement were the most accepted across cultures, while passive and active restraints were least accepted, especially in Western populations. Parental preferences varied; Jordanian parents were more accepting of passive restraint than German parents, while general anaesthesia was preferred in Bahrain. Cultural norms shaped communication styles—Latino families emphasized warm interpersonal interactions, whereas Pakistani families exhibited limited parental involvement due to language barriers. Black and Hispanic Medicaid-enrolled mothers in the U.S. reported lower satisfaction with pain management, highlighting disparities in culturally competent care. In conclusion, cultural factors significantly influence paediatric behaviour management in dental clinics. Integrating cultural competence into practice can enhance communication, improve patient compliance, and promote equitable care. Further research is needed, particularly in Africa and South America, to inform globally inclusive behaviour management guidelines.

## 1. Introduction

The behavioural management of children in a dental practice is essential for ensuring the successful delivery of oral health care and fostering positive dental experiences among them [1,2]. Various factors, including cultural norms, beliefs, and practices, shape the long-term attitudes of children toward oral healthcare [3,4]. Cultural beliefs also affect the level of parental involvement during dental visits [5] and the behaviour management techniques deemed acceptable for their children [6]. In some cultures, non-invasive techniques such as Tell–Show–Do and positive reinforcement may be preferred, while others may find techniques like voice control or protective stabilization acceptable or even necessary [7,8]. It may also inform the choice of pharmacological techniques despite the invasive nature and potential risks [8,9].

Culture may also determine the responses of the children to dental interventions [10,11]. It influences how children perceive and respond to authority figures and their tolerance for discomfort [12]. For instance, some cultures encourage stoicism in children, whereas others permit more expressive behaviours, such as crying or vocalizing distress [13,14]. These individual differences are used in tailored behaviour management techniques.

Despite the growing recognition of the influence of cultural factors on access to health care [15], there is limited evidence of its impact on behaviour management in pediatric dental settings. Guidelines and training programs for behaviour management often adopt a one-size-fits-all approach, potentially overlooking critical cultural variables that shape interactions between dentists, children, and their families [16]. It is, therefore, critical to map the existing knowledge, identify gaps, and provide actionable insights for promoting culturally competent behaviour management of children in the dental clinic. This scoping review maps the links between culture and behaviour management strategies in paediatric dental settings.

## 2. Materials and Methods

This scoping review was conducted following the Preferred Reporting Items for Systematic Reviews and Meta-Analyses Extension for Scoping Reviews (PRISMA-ScR) guidelines to ensure a transparent and systematic approach [17]. A review protocol detailing the objectives, inclusion criteria, and methodology for this scoping review was registered and published on the Open Science Framework (https://doi.org/10.17605/OSF.IO/YU87H) accessed on 4 March 2025.

### 2.1. Research Question

The scoping review was guided by the research question: How does culture influence the behaviour management of children in the dental clinic? Using the PICO framework, the target population was children receiving dental care in clinical settings (P), where the intervention involved BMTs for managing paediatric patients (I). Cultural differences in parental acceptance and preferences for BMTs were compared across diverse populations (C), with outcomes focusing on the acceptability of behaviour management strategies (O).

### 2.2. Eligibility Criteria

Articles were considered for inclusion if they were written in English, published in peer review journals, with full text available, and addressed paediatric patients, behaviour management, and cultural influence on the behaviour management of children in the dental clinic. There was no restriction by type of article, geographical location, study design, or time of publication. Other types of data sources, such as websites or books, were excluded.

### 2.3. Search Strategy

An extensive search was conducted across multiple databases and sources to identify relevant studies from the inception of the databases 31 January 2025. The search strategy utilized combinations of core concepts and keywords such as: (“behaviour management” OR “child behaviour”) AND (“cultural influence” OR “cultural factors”) AND (“paediatric dentistry” OR “dental clinic”); (“paediatric patients” OR “children”) AND (“behavioural techniques” OR “behaviour management”) AND (“cultural sensitivity” OR “cultural diversity”); (“child behaviour” AND “dental clinic”) AND (“cross-cultural” OR “cultural influence”) (Section A.1). Reference lists of the included studies, along with the grey literature, were manually reviewed to identify additional relevant studies from local, non-indexed journals. The details of the search strategy can be found in Section A.1.

### 2.4. Title, Abstract, and Article Screening

Screening was conducted independently by two reviewers (AOE and LBA) to ensure methodological rigor. The titles and abstracts of identified records were initially screened for relevance, followed by a full-text review of potentially eligible studies. In cases of disagreement regarding study inclusion, a third reviewer (MOF) served as an adjudicator to reach a consensus. Interrater reliability was assessed using the intraclass correlation coefficient (ICC) to evaluate consistency between the two primary reviewers. The agreement between reviewers was within an acceptable range, with an inter-rater reliability kappa score of 0.85, indicating a high level of agreement.

### 2.5. Data Extraction

Two authors (AOE and MOF) independently extracted the data, employing a pre-designed form. Data were extracted for the following variables: author’s name, year of publication, country, publication type, study design, sample size (male and female), study objective, behaviour management techniques influenced by culture, and key cultural findings. The included studies were appraised using expert judgment to assess the credibility and alignment of the evidence with the review objectives. Findings were presented using a table.

### 2.6. Data Analysis

A narrative summary of the extracted data was conducted. First, the summary of the characteristics of the studies included in the scoping review was conducted. Then, the extracted data on the key findings were inductively analyzed to identify themes on the influence of culture on behavior management of children in the dental clinic.

## 3. Results

The databases and sources included PubMed/MEDLINE (383 articles), Cochrane Library (102 articles), Web of Science (161 articles), and hand- search (25 articles). A total of 671 articles identified through the search strategy were downloaded into Endnote, imported into Rayyan, and 38 duplicates were removed. This left 633 articles for title and abstract screening

During the title and abstract screening phase, articles were excluded based on the following criteria: wrong publication type (85 articles), wrong population (96 articles), and wrong study outcome (167 articles). A total of 348 articles were excluded at this stage, leaving 285 articles for full-text review.

The full-text articles were assessed for eligibility based on the inclusion and exclusion criteria. More articles were excluded for the following reasons: Wrong study outcomes (199 articles); wrong population (71 articles). A total of 15 articles [18,19,20,21,22,23,24,25,26,27,28,29,30,31,32] were eligible for inclusion in the review. A PRISMA flow diagram illustrating the study selection process is shown in Figure 1.

### 3.1. Characteristics of the Included Studies

Table 1 shows that the 15 included articles were published between the years 2000 and 2024. Six of the studies were from the United States of America [18,19,20,21,24,28]; one study each from the United Kingdom [22], Australia [23], Germany and Jordan [25], Saudi Arabia [26], Spain and Portugal [27], Bahrain [29], and Sweden [29]; and two studies from India [31,32]. There were no studies from Africa and South America identified. Twelve of the included articles were original research articles, six of which are cross-sectional studies [25,27,28,29,31,32], three mixed-method studies [20,21,22], two retrospective studies [23,30], and one comparative study [19]. There was a conference paper [18], a recommendation of a diplomates’ symposium [24], and a review article [26].

The objectives of the review articles were to discuss the influence of cultural values, beliefs, practices, and language in providing dental care to diverse paediatric patients [18,22], determine factors affecting parental acceptability of behaviour management techniques (BMTs) used during dental treatment of children [19,25,27,29,30,31,32], understand the experiences that diverse families have when taking their young child to the dentist [20], describe the levels of satisfaction with dental care of children among mothers [21], assess the demand for paediatric dental general anaesthetic services [23], confront the issues of oral health disparities and the access-to-care needs of multicultural children from underserved families [24], and to review the factors that affect dental anxiety among children [26].

### 3.2. Parental Acceptance of Behaviour Management Techniques

Non-invasive techniques such as Tell–Show–Do and positive reinforcement were consistently the most accepted BMTs across different cultural contexts [19,29,31]. In contrast, more restrictive or invasive methods, including passive and active restraint, were among the least accepted, particularly in Western and European populations [25,27]. However, cultural variations were noted; for example, Jordanian parents demonstrated a greater acceptance of passive restraint compared to their German counterparts [25]. Sedation methods showed varied acceptance depending on cultural background. Conscious sedation was more accepted among non-immigrant children in Sweden [30], while Asian parents in Texas exhibited a lower acceptance of sedation compared to Caucasian and Hispanic parents [19]. In contrast, general anaesthesia was the preferred advanced BMT in Bahrain [29], whereas Jordanian parents showed a lower acceptance of this technique [25].

### 3.3. Cultural Influences on Behaviour Management Preferences

Cultural norms and social expectations seem to play a role in shaping perceptions of BMTs. For example, Latino families in the U.S. emphasize “personalismo”, a value system that promotes warm interpersonal interactions as important communication to establish personal rapport with the family. A failure to establish this rapport could lead to distrust, inaccurate history-taking, and reduced compliance with the treatment [18]. Similarly, Pakistani families in the United Kingdom exhibited a predominantly two-way communication model between the dentist and child, limiting parental involvement and support in treatment adherence [22]. Ethnicity and cultural background also influenced the perception of dental anxiety. In Arab cultures, boys were expected to exhibit bravery, whereas African cultures emphasized self-control and emotional restraint [26]. Conversely, in Western societies, children were more openly expressive of their fears and anxieties, which influenced their response to behaviour management strategies [26].

### 3.4. Sociodemographic and Linguistic Barriers

Socioeconomic status and language barriers seem to impact parental satisfaction and treatment experiences. Spanish-speaking minority families and low-income groups in the U.S. reported more negative experiences, particularly regarding the use of restraint and sedation [20]. Similarly, Black and Hispanic mothers enrolled in Medicaid were less satisfied with pain management than their White counterparts [21]. Linguistic differences also shaped communication strategies; three-way communication among White Caucasian families in the UK improved orthodontic compliance, whereas Pakistani families exhibited limited parental involvement due to language constraints [22].

## 4. Discussion

The current scoping review aimed to synthesize the existing evidence on the influence of cultural factors on behaviour management techniques used in paediatric dental settings. The findings highlight the significant role that cultural norms, beliefs, and practices play in shaping parental acceptance of BMTs, children’s responses to dental interventions, and the overall dental experience for families from diverse cultural backgrounds. The review also underscores the importance of culturally competent care in paediatric dentistry, as cultural differences can influence communication, trust, and compliance with dental treatment.

One of the strengths of the study is the diverse cultural contexts included in the review—U.S., UK, Australia, Germany, Jordan, and Saudi Arabia—that provide some insight into cultural influences on behaviour management techniques in paediatric dental settings. Also, the study followed the PRISMA guidelines, thereby ensuring a methodologically rigorous, systematic, and transparent approach to literature selection and analysis. The inclusion of diverse study designs also enriched the analysis and broadened the understanding of cultural competence in paediatric dentistry.

However, the manuscript has limitations that affect the generalizability of its findings. The geographic representation is skewed toward Western countries, with few studies from the Middle East and Asia and a notable absence of research from Africa and South America. The heterogeneity of the design, sample size, and cultural contexts included in the studies makes it difficult to draw definitive conclusions. In addition, restricting the review to English-language publications may also lead to the exclusion of relevant research in other languages, potentially limiting the diversity of perspectives. The absence of standardized definitions for key terms also introduced variability in interpretations across studies. Furthermore, the children’s age and health status can introduce potential biases due to the differences in the child’s developmental stage, cognitive ability that affects their response to BMTs, and the ability for cultural norms to shape behavioural expectations differently across age groups. Health status further affects responses to BMTs, as children with medical conditions, prior dental trauma, or high anxiety may require pharmacological interventions. Age and health status may interact with cultural factors, thereby affecting the generalizability of the conclusions. Despite these limitations, the manuscript provides a valuable contribution to the field by highlighting the importance of cultural competence in paediatric dentistry.

First, the review revealed that non-invasive techniques were consistently the most accepted BMTs across various cultural contexts. These techniques align with the principles of patient-centred care, emphasizing communication, trust-building and minimizing distress [33]. Emphasis seems to be placed on active listening, open-ended questioning, validating patient concerns, respecting patient preferences, providing clear explanations, involving family members when appropriate, and ensuring patients feel heard and understood, as observed among elderlies [34]. This essentially creates a collaborative environment where patients actively participate in their healthcare decisions [35]. These play a crucial role in shaping a child’s behaviour in the dental chair. The strategies engage specific neurological pathways that regulate fear, stress, trust, and cooperation, ultimately fostering a sense of safety and control.

Active listening and open-ended questioning engage the mirror neuron system, reinforcing empathy and reducing the likelihood of amygdala activation, mitigating fear responses [36,37]. Ensuring that children feel heard and understood keeps the brain resting while being engaged in dental care through the activation of the default mode network [38] that supports self-referential thinking and emotional processing [39]. In addition, the activity in the hypothalamicpituitary–adrenal axis is reduced, and cortisol levels decrease, leading to a calmer and more cooperative patient [40,41].

In addition, open-ended questioning stimulates the prefrontal cortex, increasing attentional control signals and reducing amygdala threat reactivity [42], thereby allowing the child to process their emotions and feel more in control [43]. The validation of concerns further supports emotional regulation by activating the anterior cingulate cortex [44], which helps dampen contextual fear responses from the amygdala while also promoting oxytocin release [45], thereby strengthening trust and reducing stress [46]. Involving family members also activates the orbitofrontal cortex, reinforcing social bonding and emotional regulation [47], enhancing oxytocin release [48], reducing stress levels [49,50], and increasing compliance.

Furthermore, respecting a child’s preferences enhances their sense of control by engaging the ventromedial prefrontal cortex [51], which is linked to decision-making and reward processing [52]. This activation promotes dopamine release, increasing motivation and cooperation [53]. Similarly, providing clear, age-appropriate explanations reduces uncertainty by engaging the hippocampus for memory formation [54] and the temporal lobe for language comprehension [55]. This process prevents the unnecessary activation of the amygdala, helping the child feel more secure [56].

These neurobiological effects of non-invasive BMTs are consistent with the broader trend in healthcare towards minimizing the use of coercive or restrictive practices in paediatric care. The lower acceptance of invasive or restrictive BMTs, particularly in Western and European populations, may be attributed to cultural values that prioritize patient autonomy, emotional well-being, and the use of evidence-based, child-centred approaches. These populations often emphasize the importance of reducing trauma and fostering positive healthcare experiences for children. In addition, the growing awareness of the long-term psychological impacts of coercive practices has led to a shift toward more empathetic and culturally sensitive care, further driving the preference for non-invasive BMTs.

The findings of the current study highlight how cultural norms and expectations can influence parental attitudes toward behaviour management. For example, we observed differences in the acceptance of passive restraint; the acceptance of conscious sedation; the communication model between dentist, child, and parent; emotional expression and coping mechanisms, which may be linked to cultural differences in pain and discomfort perception; the use of pharmacological interventions; and parent–child communication. Cultures that emphasize stoicism (restrictive emotionality and emotional regulation) and self-control [57], like the Arab and African communities, may prioritize compliance over emotional expression, whereas Western populations often favour autonomy and minimal distress [58].

Cultures significantly influence interactions in professional and healthcare settings. Collectivist cultures prioritize respect for authority [59,60,61], whereas individualistic cultures encourage independence and assertiveness [58]. These cultural orientations shape emotional expression and communication styles, with individualistic cultures fostering directness and self-reliance, while communal cultures rely on subtle cues and contextual understanding [62].

Parental cultural values also play a crucial role in shaping children’s behaviour and their interactions with authority figures, including healthcare providers. Parenting styles that emphasize obedience tend to promote compliance, enabling children to follow instructions effectively, though they may struggle with independent decision-making during care [63]. Conversely, parents who encourage negotiation foster self-expression, allowing children to actively engage with healthcare providers in the dental clinic [64].

These cultural differences can influence how children respond to behaviour management strategies [65] and the level of support they require during dental visits. Misinterpretations can arise, particularly in professional and medical settings, where cultural differences in communication can impact relationships and decision-making. These highlight the need for tailored, culturally competent approaches to ensure equitable and effective dental experiences. The need for cultural sensitivity in supporting the culture-based choices of BMTs while promoting evidence-based practices to create a supportive and reassuring environment that optimizes patient behaviour and long-term dental outcomes requires training. Poor cultural sensitivity creates barriers to service access. As indicated in the current study, families from marginalized backgrounds in the US seem to face additional challenges in accessing culturally sensitive and equitable dental care, as Black and Hispanic mothers enrolled in Medicaid were less satisfied with pain management than their White counterparts [21], and language barriers hindered the parental ability to provide support to their children during treatment [22]. This creates disparities in the quality of care received by different racial and ethnic groups.

The findings of this scoping review have several implications for paediatric dental practice. First, there is a need for dentists to adopt a culturally competent approach to behaviour management, considering the cultural norms, beliefs, and preferences of the families they serve. This may involve tailoring communication strategies, building rapport with families, and selecting BMTs that align with cultural expectations. This improves patient compliance, satisfaction, and overall dental experiences. Second, dentists should be aware of the potential impact of socioeconomic and linguistic barriers on treatment outcomes and strive to provide equitable care to all patients, regardless of their background. This may involve offering language interpretation services, providing culturally appropriate educational materials, and addressing disparities in access to care.

The included studies primarily focused on Western, Middle Eastern, and Asian populations, with limited data from African and South American contexts, restricting the generalizability of findings. Additionally, variations in methodological quality and study design may have affected the consistency and reliability of the synthesized findings. Furthermore, differences in how culture was defined and assessed across studies could have introduced inconsistencies in the interpretation of results. Finally, there is a need for further research to explore the cultural factors influencing behaviour management in paediatric dentistry, particularly in Africa and South America.

## 5. Conclusions

This scoping review highlights the significant influence of cultural factors on behaviour management in paediatric dental settings. Cultural norms, beliefs, and practices shape parental acceptance of BMTs, children’s responses to dental interventions, and the overall dental experience for families from diverse backgrounds. By adopting a culturally competent approach to care, dentists can improve communication, build trust, and ensure that all children receive equitable and effective dental treatment. Further research is needed to explore the cultural dimensions of behaviour management and to develop strategies for promoting a culturally competent use of BMTs in the paediatric dental practice.

## Figures and Tables

**Figure 1 dentistry-13-00186-f001:**
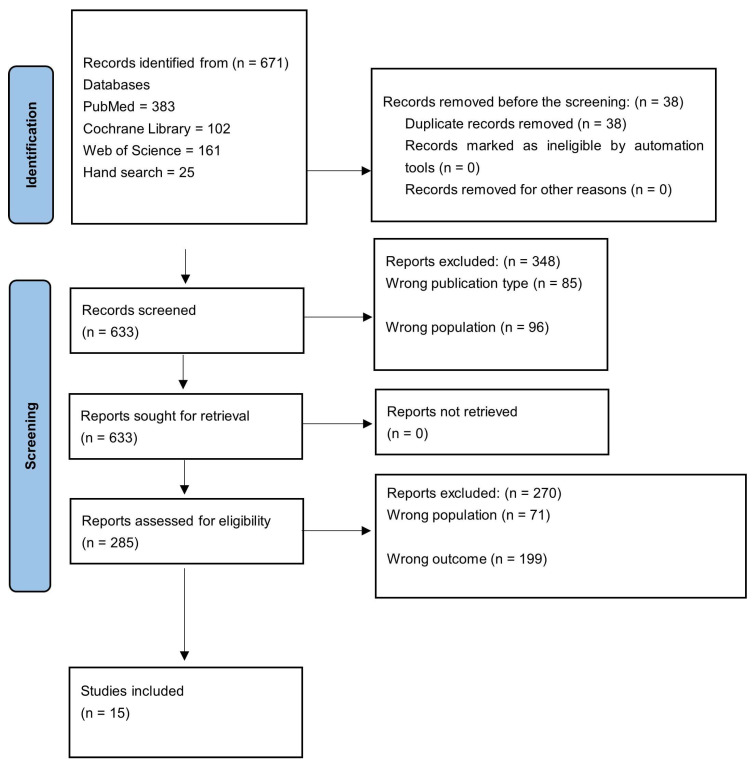
Flow diagram of the studies included in the scoping review.

**Table 1 dentistry-13-00186-t001:** Summary of characteristics of the studies included in the scoping review.

S/N	Author, Date	Country	Publication Type (Type of Study)	Sample Size (Male, Female)	Study Objective	Behaviour Management Techniques Influenced by Culture	Cultural Findings
1	Goleman, 2014 [18]	Ohio, USA	Conference paper	NA	To discuss the spectrum of cultural values, beliefs, and practices in providing dental care to diverse paediatric patients	Interpersonal interaction	Latino patients visiting their dentist would expect the ‘personalismo’, which is a formal friendliness characterized by warm relationships, close body space, handshake, hand on the shoulder or elbow, and asking about work, school, or other family members. Also, handing out a business card with your contact information would be a gesture of personalismo. If these pleasantries do not occur, the Latino family may question whether the dentist cares about them as patients, and it may lead to an inaccurate history, non-compliance, or poor follow-up.
2	Chang et al., 2018 [19]	Texas, USA	Original article (Comparative study)	104 Male: 30 Female: 74	To determine how ethnicity influences parental acceptability of behaviour management techniques used during dental treatment of children	Positive reinforcement Tell–Show–Do Voice control protective stabilization Conscious sedation	Non-invasive techniques (positive reinforcement and Tell–Show–Do) were most accepted by Caucasian, Hispanic, Asian, and African American parents, while invasive techniques (voice control and protective stabilization) were the least accepted. Asian parents were less likely to accept conscious sedation than Caucasian and Hispanic parents.
3	Reich et al., 2019 [20]	Southern California, USA	Original article (Mixed-method study)	33 Male: 0 Female: 33	To understand the experiences that diverse families have when taking their young children to the dentist and document their prevalence	Separation from caregiver Restraint Sedation	Spanish-speaking minority families and those with low incomes have negative experiences at the dentist, which seem to differ significantly from the experiences of higher income, Caucasian, and English-speaking families.
4	Milgrom et al., 2008 [21]	Washington, USA	Original article (Mixed-method study)	4191 Male: 0 Female: 4191	To describe levels of satisfaction with dental care among mothers whose preschool children are enrolled in Medicaid in Washington State	Pain management	Blacks and Hispanics were less satisfied with pain management than Whites
5	Khan &Williams, 1999 [22]	Bradford, UK	Original article (Mixed-method study)	44	To determine to what extent barriers related to culture and language and how inappropriate expectations might impede orthodontic care	Communication	Among the white Caucasian groups, three-way communication involving parents, children, and dentists enhanced understanding, supported treatment, and reinforced the need for good home care. Among Pakistani families, communication was primarily two-way, involving the dentist and the child; parents and families had a limited understanding of the process and were unable to offer a comparable level of support that would benefit their children most.
6	Alcaino et al., 2000 [23]	New South Wales, Australia	Original article (Retrospective study)	NA	To assess the demand for paediatric dental general anaesthetic services at a specialist paediatric dental unit in Australia and to evaluate the changing pattern of general anaesthetic use in children at this unit over the past decade	General anaesthesia	Whilst children of Anglo-Saxon origin accounted for the majority of patients, across the 13 years, there was a significant increase in the number of children from Asian or Middle Eastern backgrounds. However, over 50% of the Asian and Middle-Eastern children accessed the GA service through the emergency department. This suggests that they were motivated to seek treatment by pain rather than an expectation of comprehensive care.
7	Punwani, 2003 [24]	Chicago, IL, USA	Recommendations of Diplomates symposium	NA	To confront the issues of oral health disparities and the access-to-care needs of multicultural children from underserved families	Communication	The chances for miscommunication may increase when providers care for patients from other cultures who may have a less well-organized conceptualization of illness or who may face particularly difficult social conditions.
8	Al Zoubi et al., 2021 [25]	Germany and Jordan	Original article (Cross-sectional study)	100 Male: 24 Female: 76	To investigate the differences in parental acceptance of advanced behaviour management techniques in different cultural backgrounds (Germany vs. Jordan)	Passive restraint Active restraint Nitrous oxide sedation General anaesthesia	Nitrous oxide sedation was the most accepted advanced behavioural management technique. The least acceptable technique in Germany was passive restraint and, in Jordan, general anaesthesia. The parents in Germany are significantly more accepting of nitrous oxide sedation than are parents in Jordan, while parents in Jordan are more willing to accept passive restraint. The acceptance of all advanced BMT increased significantly in both groups when the treatment was urgent.
9	Alasmari et al., 2018 [26]	Saudi Arabia	Review article	NA	To review the factors that affect dental anxiety and provide an insight into the possible explanations on the influence of these factors	Level of dental anxiety	Ethnicity and cultural background may influence the level of dental anxiety. In the Arab cultural background, it has been suggested that boys are expected to act like men and to be brave. On the other hand, in African culture, endurance to stress usually manifests as self-control and self-repression. However, in American or European cultures, children can more easily express their anxiety and feelings.
10	Guinot et al., 2021 [27]	Spain and Portugal	Original article (Cross-sectional study)	100 Male: 33 Female: 67	To compare the acceptance of behaviour management techniques used in paediatric dentistry by Spanish and Portuguese parents	Tell–Show–Do, Voice control Nitrous oxide sedation Oral premedication General anaesthesia Hand over mouth Active restraint Passive restraint	Tell–Show–Do and voice control were rated the most acceptable techniques in both Spain and Portugal, whereas the least accepted techniques in both countries were active and passive restraint.
11	Hill et al., 2019 [28]	Chicago, IL, USA	Original article (Cross-sectional study)	266	To determine if caregivers’ race and ethnicity impact their willingness to accept passive immobilization for their child’s dental treatment	Passive immobilization	Caregiver race/ethnicity impacts their willingness to accept passive immobilization. Hispanics (84%) were more willing than African Americans (66%), Asians (50%), and Caucasians (24%).
12	Abdulla et al., 2024 [29]	Bahrain	Original article (Cross-sectional study)	140	To investigate parental acceptance of several BGT in some private and governmental dental clinics in Bahrain	Tell–Show–Do Tell–Play–Do Direct observation 3-D distraction Positive reinforcement Voice control Hand Over mouth Exercise Parental presence/absence Protective stabilization N_2_O/O_2_ sedation General anaesthesia	Positive reinforcement followed by TellShow–Do and TellPlay–Do were the most accepted by Bahraini parents. General anaesthesia was the most accepted advanced BMT, followed by protective stabilization, N_2_O/O_2_ sedation, parental absence, and voice control.
13	Dahlander et al., 2015 [30]	Sweden	Original article (Retrospective study)	223	To study the choice of sedation method among children with immigrant backgrounds	Conscious sedation	Conscious sedation was used significantly more often in the non-immigrant group.
14	Acharya, 2017 [31]	India	Original article (Cross-sectional study)	50 Male: 32 Female: 18	To assess the parents’ acceptance of the behaviour management techniques commonly used in paediatric dentistry.	Voice control Tell–Show–Do Positive Reinforcement Mouth prop Modelling Hand over mouth Exercise (HOME) Physical restraint Oral premedication N_2_O/O_2_ sedation General anaesthesia	The Tell–Show–Do technique was the most accepted behaviour technique, and hand over mouth exercise was the least accepted behaviour technique.
15	Shukla et al., 2021 [32]	India	Original article (Cross-sectional study)	50	To evaluate parental acceptance towards behaviour management techniques at the side of its reference to previous dental expertise and dental anxiety	Tell–Show–Do Voice control Modelling Hand over mouth Exercise Active restraint Parental presence/absence Audiovisual Oral sedation General anaesthesia	The most accepted technique was the audiovisual technique followed by Tell–Show–Do and anaesthesia. The least accepted technique was oral sedation.

## Data Availability

All the data used for this study are publicly accessible.

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
