# Peer review of "Culture and Behaviour Management of Children in the Dental Clinic: A Scoping Review"

_dentistry, 2025, doi:10.3390/dj13050186_

Round 1
Reviewer 1 Report
Comments and Suggestions for Authors
I would like to thank you for the opportunity to review this scoping review, which explores a topic of great interest to pediatric dentists: the impact of culture on behaviour management strategies in pediatric dental settings.
Below are some suggestions that I hope will be helpful in improving the paper.
- Use of PRISMA-ScR: The PRISMA-ScR guidelines should be followed instead of the standard PRISMA checklist, which is designed for systematic reviews. Please ensure full compliance with PRISMA-ScR and include the completed checklist for transparency.
- Reference 17: This reference requires revision—please verify and ensure alignment with the correct checklist.
- Eligibility Criteria: The section lacks details on the types of articles included. Additionally, the screening process is unclear, particularly regarding exclusion criteria. Please clarify these aspects.
- Figure Placement: Figure 1, which presents research results, is misplaced. It should be relocated to the Results section to improve logical flow.
- Missing Data Analysis: The manuscript does not provide a clear explanation of the data analysis process. Please add a dedicated paragraph detailing the methodology.
- Reference 34: This reference focuses on elderly populations rather than children, making it unsuitable for this study. Please replace it with a more relevant source.
- Discussion Section: The discussion primarily addresses physiological processes, with insufficient focus on the cultural factors influencing parental acceptance. A stronger emphasis on cultural impact is needed to align with the study’s objectives.
- Reference Revisions: Please review and update all references throughout the manuscript to ensure accuracy and relevance.
Author Response
I would like to thank you for the opportunity to review this scoping review, which explores a topic of great interest to pediatric dentists: the impact of culture on behaviour management strategies in pediatric dental settings. Below are some suggestions that I hope will be helpful in improving the paper.
Response: Thanks for the constructive feedback. Below are the point-by-point responses to the comments.
- Use of PRISMA-ScR: The PRISMA-ScR guidelines should be followed instead of the standard PRISMA checklist, which is designed for systematic reviews. Please ensure full compliance with PRISMA-ScR and include the completed checklist for transparency.
Response: Thanks for identifying this error. The correction has been made and the correct guideline attached.
- Reference 17: This reference requires revision—please verify and ensure alignment with the correct checklist.
- Response: Thanks for raising the concern. The reference statement has been revised to read: This scoping review was conducted following the Preferred Reporting Items for Systematic Reviews and Meta-Analyses Extension for Scoping Reviews guidelines (PRISMA-ScR) to ensure a transparent and systematic approach Tricco AC, Lillie E, Zarin W, O’Brien KK, Colquhoun H, Levac D, et al. PRISMA Extension for scoping reviews (PRISMAScR): Checklist and Explanation. Ann Intern Med. 2018;169:467–73.
- Eligibility Criteria: The section lacks details on the types of articles included. Additionally, the screening process is unclear, particularly regarding exclusion criteria. Please clarify these aspects.
Response: Thanks for highlighting this gap. We agree with the reviewer and have rewritten the section. We wrote: Articles were considered for inclusion if they were written in English, published in peer review journals, with full text available, and addressing paediatric patients (children), behaviour management, and cultural influence on the behaviour management of children in the dental clinic. There was no restriction by type of article, geographical location, study design or time of publication. Other types of data sources such as websites or books were excluded.
- Figure Placement: Figure 1, which presents research results, is misplaced. It should be relocated to the Results section to improve logical flow.
Response: Thanks. This has been correctly placed in the result section along with the narrative.
- Missing Data Analysis: The manuscript does not provide a clear explanation of the data analysis process. Please add a dedicated paragraph detailing the methodology.
Response. Thanks for highlighting this gap. We wrote: A narrative summary of the extracted data was done. First, the summary of the characteristics of the studies included in the scoping review was done. Then the extracted data on the key findings were inductively analyzed to identify themes on the influence of culture on behavior management of children in the dental clinic.
- Reference 34: This reference focuses on elderly populations rather than children, making it unsuitable for this study. Please replace it with a more relevant source.
Response: Thanks for highlighting this. We retained the reference and modified the sentence; Emphasis seems to be placed on active listening, open-ended questioning, validating patient concerns, respecting patient preferences, providing clear explanations, involving family members when appropriate, and ensuring patients feel heard and understood as observed among elderlies [34].
- Discussion Section: The discussion primarily addresses physiological processes, with insufficient focus on the cultural factors influencing parental acceptance. A stronger emphasis on cultural impact is needed to align with the study’s objectives.
Response: Thanks for raising this. We have strengthened the section by including this discussion on culture: Cultures that emphasizes stoicism (restrictive emotionality and emotional regulation) and self-control [57] like the Arab and African communities, may prioritize compliance over emotional expression, whereas Western populations often favour autonomy and minimal distress [58].
Cultures significantly influence interactions in professional and healthcare settings. Collectivist cultures prioritize respect for authority [59-61], whereas individualistic cultures encourage independence and assertiveness [58]. These cultural orientations shape emotional expression and communication styles, with individualistic cultures fostering directness and self-reliance, while communal cultures rely on subtle cues and contextual understanding [62].
Parental cultural values also play a crucial role in shaping children's behaviour and their interactions with authority figures, including healthcare providers. Parenting styles that emphasize obedience tend to promote compliance, enabling children to follow instructions effectively, though they may struggle with independent decision-making during care [63]. Conversely, parents who encourage negotiation foster self-expression, allowing children to actively engage with healthcare providers in the dental clinic [64].
- Reference Revisions: Please review and update all references throughout the manuscript to ensure accuracy and relevance.
Response: Thanks for the advice. Done.
Reviewer 2 Report
Comments and Suggestions for Authors
An interesting study and one that underscores the importance of cultural mores into managing children in a dental environment.
The mean question of this research is Does culture influence the type of behavior management techniques. This study was a nice review of behavior management acceptable to parents as influenced by cultural mores. This filled a gap in the field and was a nice addition to the literature. It is a summary of multiple studies and the conclusions were consistent with the methodology. The references, tables and figures are all appropriate.
Author Response
Reviewer II
An interesting study and one that underscores the importance of cultural mores into managing children in a dental environment.
Response: Thanks for the constructive feedback. This has helped improve the quality of the manuscript.
The mean question of this research is Does culture influence the type of behavior management techniques. This study was a nice review of behavior management acceptable to parents as influenced by cultural mores. This filled a gap in the field and was a nice addition to the literature. It is a summary of multiple studies and the conclusions were consistent with the methodology. The references, tables and figures are all appropriate.
Response: Thanks once again for the constructive feedback.
Reviewer 3 Report
Comments and Suggestions for Authors
Title Comments: The PRISMA guideline suggests including “systematic review” in the manuscript title to clearly identify the report.
Abstract Comments:
- Background: To better evaluate the impact, it is suggested to clarify that the objective was to identify or determine the cultural influence on the behavior management of children.
- Methods: Consider specifying the period during which the included articles were published.
- Results: The methods used to present and synthesize results were not mentioned.
- Conclusion: Although a general interpretation of the results is provided, there is no mention of study risk of bias, inconsistency, or imprecision.
- Key word: words and phrases that suggest what the topic is about. It’s not the case of “non-pharmacological”
Methods:
- Delivering oral health care in a regular dental chair during childhood is always a challenge. However, the authors did not mention two key components regarding this population: the children's age and whether they were all healthy patients (potential study bias).
Eligibility Criteria:
- It is suggested that the authors consider using the PICO framework to organize and facilitate the reader’s understanding.
Data Analysis:
- The authors do not refer to any analysis measures (e.g., assessing the association between behaviors and gender, age, cultural background).
Results:
- Characteristics of Included Studies:
- There is an error in line 140. Is the sample size n=15 or 16? Additionally, Figure 1 is incorrectly referred to as a table.
- Reference #20 was cited as a 2020 publication instead of 2019.
- Table 1: It is suggested that the authors summarize the column regarding “cultural findings.”
"It would be beneficial to enhance the English writing throughout the manuscript for better clarity and readability."
Author Response
Reviewer III
Title Comments: The PRISMA guideline suggests including “systematic review” in the manuscript title to clearly identify the report.
Response: Thanks for the advice. We conducted a scoping review. This is in the title. We have also changed the checklist to the appropriate checklist.
Abstract Comments:
- Background: To better evaluate the impact, it is suggested to clarify that the objective was to identify or determine the cultural influence on the behavior management of children.
Response: Thanks for the suggestion. We acknowledge that this suggestion is very apt for a systematic review. For a scoping review however, the objective is to map the literature and we have done that with the current review. For this reason, we have edited the objective to be more reflective of the scope of the review. We wrote: This scoping review maps the links between culture and behaviour management strategies in paediatric dental settings.
- Methods: Consider specifying the period during which the included articles were published.
Response: Thanks for the suggestion. This was a major oversight. We wrote: … from the inception of the data base till January 31 2025.
- Results: The methods used to present and synthesize results were not mentioned.
Response: Thanks for highlighting this gap. We wrote: Data on the key findings were inductively analyzed to ….
- Conclusion: Although a general interpretation of the results is provided, there is no mention of study risk of bias, inconsistency, or imprecision.
Response: Thank you for highlighting this concern. As this study employed a scoping review methodology rather than a systematic review, formal assessments of risk of bias, inconsistency, or imprecision were not conducted, as such evaluations are not typically required for scoping reviews (Tricco et al., 2018). This citation from the PRISMA-ScR guidelines explicitly states that scoping reviews prioritize mapping evidence breadth rather than appraising quality, which justifies the absence of bias assessment.
- Key word: words and phrases that suggest what the pic is about. It’s not the case of “non-pharmacological”
Response: Thanks for the guidance. We have now included two synonyms - Parenting styles; Stoicism
Methods:
- Delivering oral health care in a regular dental chair during childhood is always a challenge. However, the authors did not mention two key components regarding this population: the children's age and whether they were all healthy patients (potential study bias).
Response: Making the distinction between children's ages and their health status is challenging. Since the study includes a mix of reviews, opinion pieces, and parent-reported data, not all manuscripts provided specific details on these factors. We acknowledge the potential for study bias and have highlighted this as a limitation in our discussion. We wrote: In addition, the children's age and health status can introduce potential biases due to the differences in the child's developmental stage, cognitive ability that affects their response to BMTs and the ability for cultural norms to shape behavioral expectations differently across age groups. Health status further affects responses to BMTs, as children with medical conditions, prior dental trauma, or high anxiety may require pharmacological interventions. Age and health status may interact with cultural factors thereby affecting the generalizability of the conclusions.
Eligibility Criteria:
- It is suggested that the authors consider using the PICO framework to organize and facilitate the reader’s understanding.
Response: Thanks for the guidance. We wrote: Using the PICO framework, "The target population was children receiving dental care in clinical settings (P), where the intervention involved BMTs for managing pediatric patients (I). Cultural differences in parental acceptance and preferences for BMTs were compared across diverse populations (C), with outcomes focusing on the acceptability of behavior management strategies (O).
Data Analysis:
- The authors do not refer to any analysis measures (e.g., assessing the association between behaviors and gender, age, cultural background).
Response: We wrote: A narrative summary of the extracted data was done. First, the summary of the characteristics of the studies included in the scoping review was done. Then the extracted data on the key findings were inductively analyzed to identify themes on the influence of culture on behavior management of children in the dental clinic.
Results:
Characteristics of Included Studies:
- There is an error in line 140. Is the sample size n=15 or 16? Additionally, Figure 1 is incorrectly referred to as a table.
Response: Thank you for identifying this error. The number has been corrected to 15. Additionally, the figure has been appropriately referenced
- Reference #20 was cited as a 2020 publication instead of 2019.
Response: Apologies, we could not identify this error.
- Table 1: It is suggested that the authors summarize the column regarding “cultural findings.”
Response: Long narrative were shortened.
Comments on the Quality of English Language
"It would be beneficial to enhance the English writing throughout the manuscript for better clarity and readability."
Response: Thanks for the suggestion. A grammar check was done.
Round 2
Reviewer 1 Report
Comments and Suggestions for Authors
I commend the authors for their efforts in improving the paper, which I now consider suitable for publication in its current form
Reviewer 3 Report
Comments and Suggestions for Authors
No more comments.